# Design of a Cryogenic Duplex Pressure-Swirl Atomizer through CFDs for the Cold Conservation of Marine Products

Eduardo Ayala [1], Diego Rivera [1,*], Julio Ronceros [1,*], Nikolai Vinces [1] and Gustavo Ronceros [2]

1  Mechatronics Department, Universidad Peruana de Ciencias Aplicadas, Santiago de Surco, Lima 15023, Peru; u201411982@upc.edu.pe (E.A.); leonardo.vinces@upc.edu.pe (N.V.)

2  Instituto Latino-Americano de Tecnologia e Infraestrutura, Universidade Federal de Integração Latino—Americana (UNILA), Foz do Iguaçu 85870-650, PR, Brazil; gustavo_ronceros@hotmail.com

*  Correspondence: u201414022@upc.edu.pe (D.R.); julio.ronceros@upc.pe (J.R.)

**Abstract:** The following article proposes the design of a bi-centrifugal atomizer that allows the interaction of sprays from two fluids (water and liquid nitrogen). The liquid nitrogen ($LN_2$) is below $-195.8$ °C, a temperature low enough for the nitrogen, upon contact with the atomized water, to cause heat loss and bring it to its freezing point. The objective is to convert the water droplets present in the spray into ice. Upon falling, the ice particles can be dispersed, covering the largest possible area of the seafood products intended for cold preservation. All these phenomena related to the interaction of two fluids and heat exchange are due to the bi-centrifugal atomizer, which positions the two centrifugal atomizers concentrically, resulting in the inevitable collision of the two sprays. Each of these atomizers will be designed using a mathematical model and CFDs tools. The latter will provide a better study of the flow behavior of both fluids inside and outside the bi-centrifugal atomizer. Hence, the objective revolves around confirming the validity of the mathematical model through a comparison with numerical simulation data. This comparison establishes a strong correlation (with a maximum variance of 1.94% for the water atomizer and 10% for the $LN_2$ atomizer), thereby ensuring precise manufacturing specifications for the atomizers. It is important to highlight that, in order to achieve the enhanced resolution and comprehension of the fluid both inside and outside the duplex atomizer, two types of meshes were utilized, ensuring the utilization of the optimal option. Similarly, the aforementioned meshes were generated using two distinct software platforms, namely ANSYS Meshing (tetrahedral mesh) and ANSYS ICEM (hexahedral mesh), to facilitate a comparative analysis of the mesh quality obtained. This comprehension facilitated the observation of water temperature during its interaction with liquid nitrogen, ultimately ensuring the freezing of water droplets at the atomizer's outlet. This objective aligns seamlessly with the primary goal of this study, which revolves around the preservation of seafood products through cold techniques. This particular attribute holds potential for various applications, including cooling processes for food products.

**Keywords:** Ansys Fluent; CFD; Abramovich theory; Kliachko theory; multiphase VOF; cryogenic atomizer; liquid nitrogen; open-end pressure-swirl atomizer; closed pressure-swirl atomizer

## 1. Introduction

This research endeavors to delve into the meticulous examination and enhancement of preservation methodologies within the realm of Peru's fishing industry. An industry that has adeptly harnessed the creation of legitimate employment, financial gains for the Peruvian government, and proficient export practices, thereby engendering a pivotal contribution to the decentralized growth of the nation's economy. This sector, which occupies a paramount standing within the economic framework, stands amongst the four preeminent contributors to the country's foreign exchange earnings. The Central Reserve Bank of Peru substantiates that the fishing sector contributes a noteworthy 7% to the overall national export figures [1]. Despite these commendable achievements, the sector encounters

intricate challenges arising from the intrinsic variability of vessels and fishing volumes. These challenges necessitate a diverse array of refrigeration techniques, an initial investment that can prove financially burdensome, and a subsequent long-term maintenance commitment, further compounded by their comparatively elevated energy consumption. This confluence of factors often steers many vessels towards adopting less efficient and budget-conscious systems. For instance, the prevalent utilization of ice machines to sustain the integrity of catches during maritime sojourns is a common practice. When faced with these multifaceted challenges, an innovative approach emerged: the potential integration of a pioneering cold preservation method, harnessing the distinctive attributes of cryogenic fluids, prominently liquid nitrogen ($LN_2$), characterized by a remarkably low boiling point of $-195.8\ ^\circ$C [2]. The exploration undertaken in this research extends to the meticulous evaluation and viability assessment of this novel preservation technique.

This investigation capitalizes on the efficacy of dual or bi-centrifugal atomization mechanisms, denoted as duplex atomizers, employing centrifugal or pressure-swirl atomizers [3]. The discernible conical geometry exhibited by the resultant spray from these atomizers facilitates immediate interaction between the atomized substances [4,5]. Consequently, this dynamic interaction triggers the dissipation of heat from water droplets, facilitated by the presence of liquid nitrogen particles [6–8]. In its essence, this research converges upon the central objective of addressing the multifaceted preservation challenges encountered within the sphere of the fishing industry. This endeavor inherently unfurls avenues for novel perspectives, particularly pertaining to the proficient cooling of comestible products and the innovation of novel industrial applications.

## 2. Mathematical Model

In this project, a bi-centrifugal atomizer model with tangential channels [9,10] will be used. In order to achieve this, input parameters are defined, which, when used in the equations of the mathematical models, will provide us with the main dimensions of the centrifugal atomizers. The mathematical model to be employed consists of two stages: in the first stage, the fluid is considered inviscid, and then in the final stage, viscosity losses are taken into account [11]. These stages are show in Figure 1.



**Figure 1.** Methodology block diagram.

This design method was selected due to the nonmeasurable and challenging-to-model nature of the air core radius. Therefore, an approach is sought that transforms everything into a function of the annular section coefficient ($\varphi$) [12]. As a result, the decision was made to utilize a conical model as a foundation, as depicted in the article "Development of a mathematical model and 3D numerical simulation of the internal flow in a conical swirl atomizer" [3]. In this context, Equation (2) is a derivation of the conical geometric parameter equation ($A_c$) (Equation (1)) [3].

$$A_c = \frac{\pi r_o R_{inj}}{n f_p} \cos \psi \sin \beta \dots \dots \tag{1}$$

Given its tangential model nature, it is considered that ($\psi = 0^\circ$ y $\beta = 90^\circ$), which results in a method with which to determine the geometric parameter (A) with measurable

variables (Equation (2)). This equation is then transformed into a function of the annular section coefficient (Equation (4)):

$$A = \frac{\pi r_o R_{inj}}{n f_p} \ldots\ldots \tag{2}$$

Firstly, the initial design parameters of the atomizer need to be defined. These parameters are as follows: they are the design parameters for the required operation of the atomizer and are required for the injection system that supports the pressure of n channels with a spray angle; these parameters are defined at the beginning and by using reverse engineering for the atomizer geometry.

Additionally, the physical properties of each fluid must be obtained, with the main properties as follows:

Once the initial parameters (Table 1) and fluid properties (Table 2) are defined, the next step is to determine the parameters by assuming an ideal liquid. We begin by calculating the annular section coefficient "φ" by using Equation (3), where the annular section coefficient is derived from the spray semi-angle.

$$\sin\alpha \cong \frac{2\sqrt{2}(1 - \varphi)}{\left(1 + \sqrt{1 - \varphi}\right)\sqrt{2 - \varphi}} \ldots\ldots \tag{3}$$

**Table 1.** Initials parameters.

| Initial Parameters | Water | LN$_2$ |
|---|---|---|
| Spray Angle, $2\alpha$ (°) | 130 | 135 |
| Mass Flow Rate, m (kg. s$^{-1}$) | 0.245 | 0.035 |
| Pressure differential, $\Delta P$ (kPa) | 350 | 350 |
| Number of channels, "n" | 4 | 4 |

**Table 2.** Physical properties.

| Physics Properties of Fluid | Water | LN$_2$ |
|---|---|---|
| Density, $\rho$, (kg.m$^{-3}$) | 1000 | 806.08 |
| Kinematic viscosity, $\nu$ (m$^2$.s$^{-1}$) | $10^{-6}$ | $19 \times 10^{-8}$ |
| Absolut viscosity, $\mu$ (kg. (m.-s)$^{-1}$) | $1.003 \times 10^{-3}$ | $160.65 \times 10^{-6}$ |
| Surface tension, $\sigma$, (N. m$^{-1}$) | 0.072 | 0.0263 |

Once the annular section coefficient is determined, the constructive geometric parameter of the injector "A" and its corresponding discharge coefficient "Cd" can be calculated using Equations (4) and (5), respectively.

$$A = \frac{\sqrt{2}(1 - \varphi)}{\varphi\sqrt{\varphi}} \ldots\ldots \tag{4}$$

$$C_d = \sqrt{\frac{\varphi^3}{2 - \varphi}} \ldots\ldots \tag{5}$$

With these obtained data, the radius of the injector discharge orifice "r$_o$" can be calculated using Equation (6):

$$C_d = \frac{\dot{m}}{\pi r_o^2 \sqrt{2\rho\Delta P}} \ldots \tag{6}$$

Next, Equation (7) relates the swirl chamber radius "Rs" to the radius of the tangential channel "r$_{inj}$" (Figure 2). Additionally, Equation (8) relates the main diameters of the

atomizer using the constructive geometric parameter "A", considering that the atomizers are open (C = 1) [13], which means that "Rs = r$_o$".

$$R_{inj} = R_S - r_{inj} \dots\dots\dots\dots \tag{7}$$

$$A = \frac{r_o R_{inj}}{n.r_{inj}^2} \dots\dots\dots\dots\dots \tag{8}$$

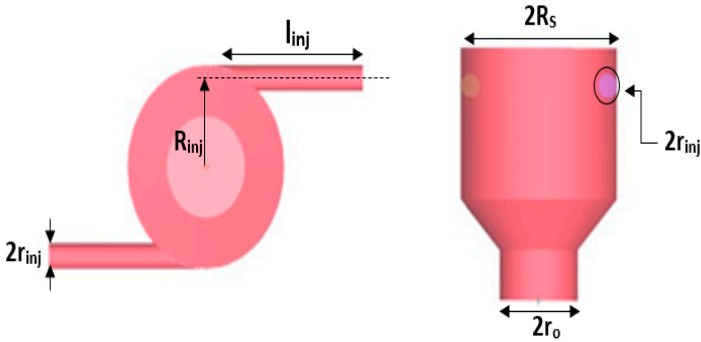

**Figure 2.** Main dimensions of the centrifugal atomizer.

Finally, by inserting Equation (7) into Equation (8), Equation (9) is obtained, which allows for the calculation of the radius of the tangential channel "r$_{inj}$".

$$r_{inj}^2 + \frac{R_s}{A.n}.r_{inj} - \frac{R_s^2}{A.n} = 0 \dots\dots \tag{9}$$

With the obtained parameters, the length of each tangential channel can be defined, taking into consideration the following:

$$l_{ent} = 15r_{inj} \dots\dots\dots\dots\dots \tag{10}$$

Next, the effects of viscosity are taken into account, where, in the first instance, Equation (11) is used to determine the inlet velocity in each channel of the atomizer, which is related to the Reynolds number (Re) and the Blasius coefficient ($\lambda = 0.3164\text{Re}^{-1/8}$) [14,15].

$$U_{ent} = \frac{\dot{m}}{\rho n \pi r_{inj}^2} \dots \tag{11}$$

For an open atomizer, the equivalent geometric parameter "A$_{eq}$" is equal to "A" since, in this case, K = 1 (loss coefficient due to the decrease in angular momentum). Then, by applying Equation (12), the annular section coefficient "$\varphi_{eq}$" is obtained, which is important for calculating the equivalent discharge coefficient "Cd$_{eq}$" (Equation (13)) and the equivalent spray semi-angle "$\alpha_{eq}$" (Equation (14)).

$$A_{eq} = AK = \frac{R_{inj}r_o}{nr_{inj}^2 + \frac{\lambda}{2}R_{inj}(R_s - r_o)} = \frac{\sqrt{2}(1 - \varphi_{eq})}{\varphi_{eq}\sqrt{\varphi_{eq}}} \dots \tag{12}$$

$$Cd_{eq} = \frac{1}{\sqrt{\frac{2 - \varphi_{eq}}{\varphi_{eq}^3} + \xi_{tot}\left(\frac{Ar_o}{R_{inj}}\right)^2}} \dots \tag{13}$$

$$\sin\alpha_{eq} = \frac{2Cd_{eq}A_{eq}}{(1 + \sqrt{1 - \varphi_{eq}})\sqrt{1 - \xi_{tot}Cd_{eq}^2\left(\frac{Ar_o}{R_{inj}}\right)^2}} \dots \tag{14}$$

Finally, by applying Equation (6) again in terms of the equivalent discharge coefficient "$Cd_{eq}$", the corrected radius of the outlet orifice "$r'_o$" (Equation (15)) is obtained. Then, by applying the aforementioned equations, the corrected dimensions of the atomizer are obtained. The tables below show the dimensions obtained using the inviscid model and the viscous model [16] for both the LN$_2$ atomizer (Table 3) and the water atomizer (Table 4).

$$r'_o = \sqrt{\frac{\dot{m}}{\pi . Cd_{eq}\sqrt{2\rho\Delta P}}} \cdots \cdots \tag{15}$$

**Table 3.** Parameters of the LN$_2$ atomizer.

| Parameters | Ideal Liquid | Losses Viscosity |
|---|---|---|
| $\varphi$ | 0.125 | 0.125 |
| A | 27.85 | 27.85 |
| cd | 0.0324 | 0.0237 |
| $R_s$ | 3.802 | 4.45 |
| $r_o$ | 3.802 | 4.45 |
| $r_{inj}$ | 0.477 | 0.552 |
| $l_{ent}$ | 4.765 | 5.574 |

**Table 4.** Parameters of the water (H$_2$O) atomizer.

| Parameters | Ideal Liquid | Losses Viscosity |
|---|---|---|
| $\varphi$ | 0.173 | 0.173 |
| A | 16.258 | 16.258 |
| cd | 0.05323 | 0.0392 |
| $R_s$ | 7.445 | 8.672 |
| $r_o$ | 7.445 | 8.672 |
| $r_{inj}$ | 1.196 | 1.393 |
| $l_{ent}$ | 11.962 | 13.932 |

## 3. Numerical Simulation

By using the dimensions obtained from the mathematical model, the bi-centrifugal atomizer was designed using the ANSYS SPACECLAIM software (Figure 3). In the upper part, the nitrogen atomizer with its four tangential channels can be visualized, and in the lower part, with a larger diameter and located concentrically, the water atomizer with its respective inlet channels can be seen.

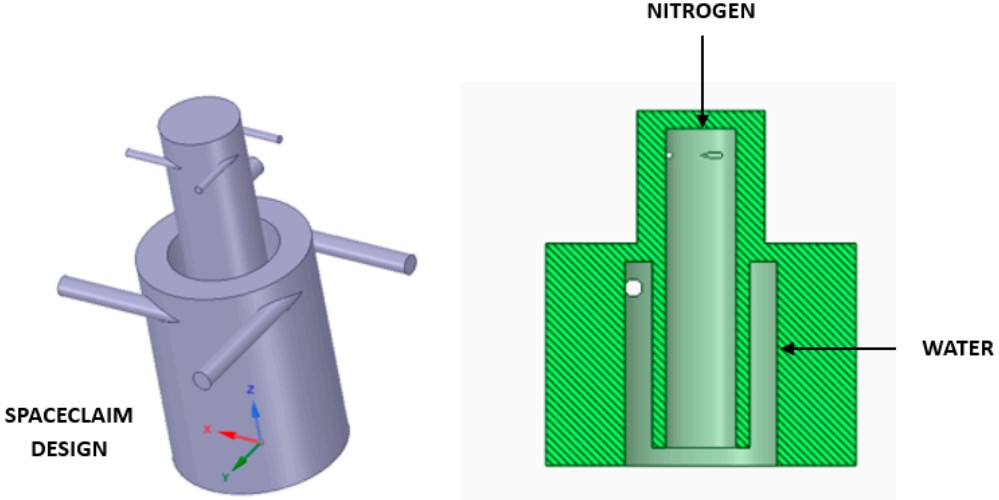

**Figure 3. Left side**: isometric view of the atomizer; **right side**: section view of the atomizer.

Within the domain of computational fluid dynamics (CFD), the orchestration of meshing emerges as a paramount endeavor in the numerical scrutiny of fluid dynamics. Of noteworthy significance are the two principal classifications of meshes: the tetrahedral and hexahedral variants. Each configuration presents a distinctive array of merits and demerits, thereby underscoring the profound significance of the choice between the two. This decision is contingent upon the intricate inclinations of the simulation at hand.

Subsequently, the mesh for the bi-centrifugal atomizer was fashioned utilizing the Ansys Meshing software. It is imperative to underscore that both atomizers were concentrically amalgamated, leading to the generation of a tetrahedral mesh [17]. Moreover, to refine the mesh design, a hexahedral mesh was engendered for the purpose of juxtaposing and scrutinizing simulation outcomes. (Figures 4–6).

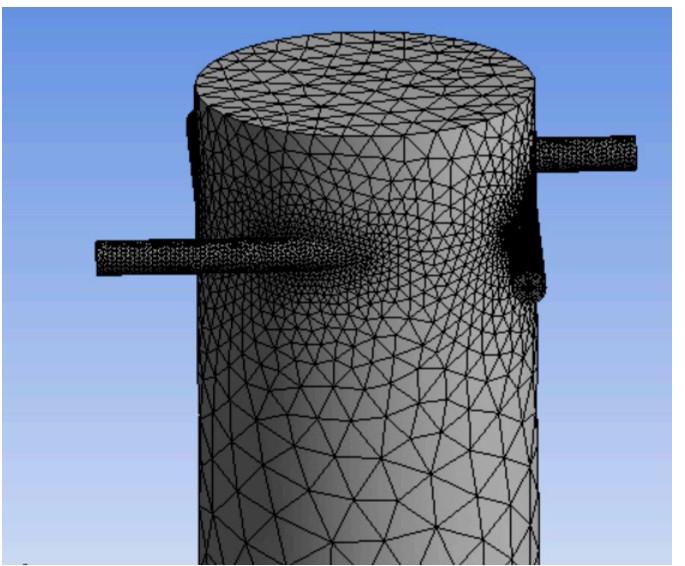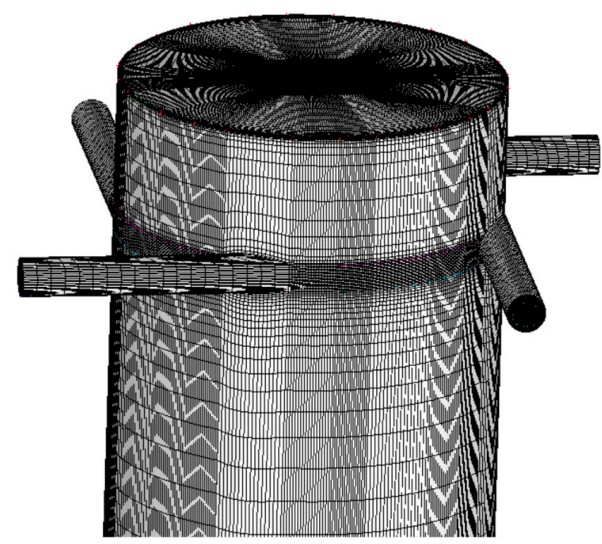

**Figure 4.** Mesh of nitrogen atomizer—**left side**: tetrahedral; **right side**: hexahedral.

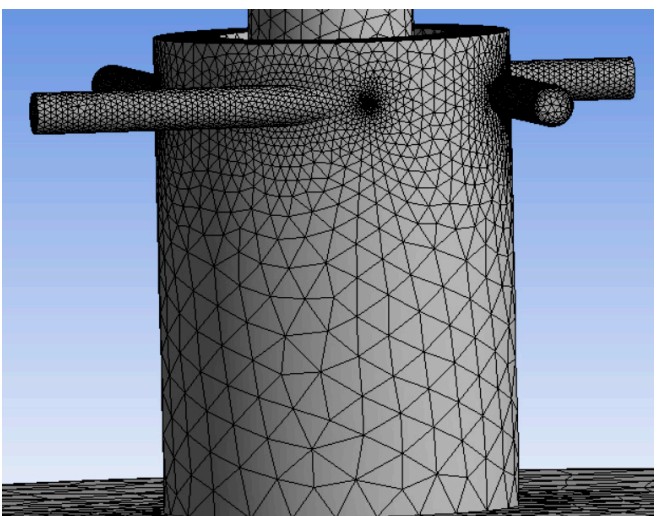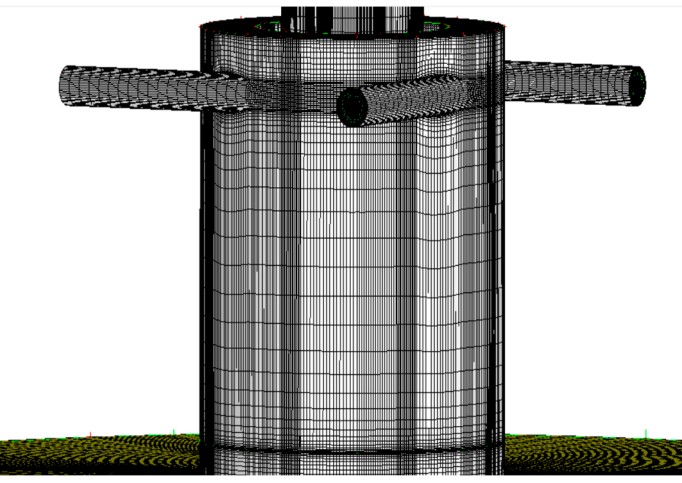

**Figure 5.** Mesh of water atomizer—**left side**: tetrahedral; **right side**: hexahedral.

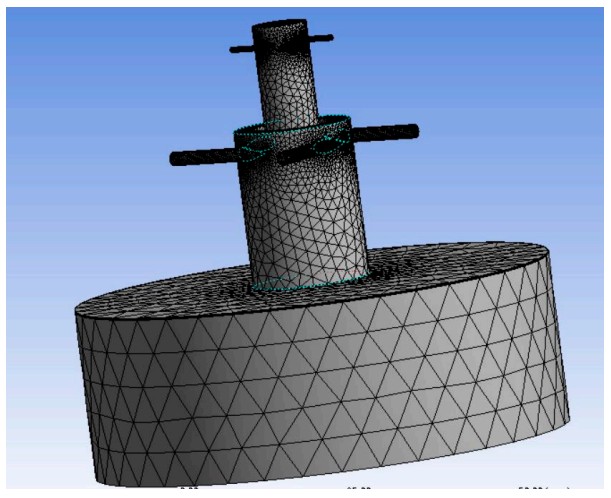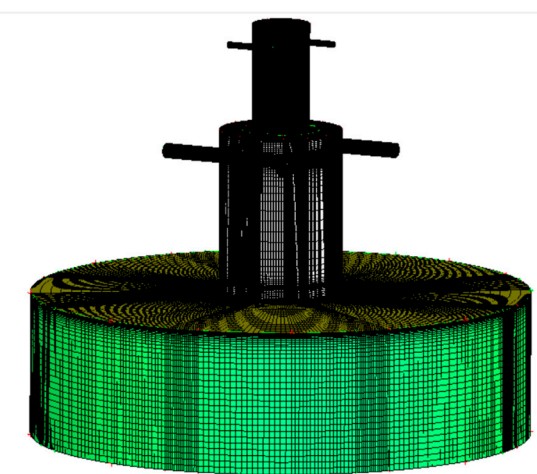

**Figure 6.** Mesh of duplex atomizer—**left side**: tetrahedral; **right side**: hexahedral.

In the numerical simulation, a tetrahedral and hexahedral mesh were used to control the internal flow of both atomizers, taking into account the inlet boundary conditions with a specific pressure, as well as a specific outlet pressure and no slip on the walls [18]. Subsequently, in order to achieve mesh independence in the simulation, it is necessary to consider the dimensionless distance from the wall, which is known as (**y**+). Based on this dimensionless parameter, it is possible to identify the appropriate region to address turbulence-related phenomena, as indicated by Equation (16).

$$y^+ = \frac{\rho U_\tau y}{\mu} \dots \tag{16}$$

Then, by considering a range of speeds of the free flow (U∞) and a hydraulic diameter for the inlet channels (Dh = 1 mm), in the meshes made, a skin coefficient "Cf" can be obtained, which is used in Equation (17) [19]. By calculating the shear stress of the wall ($\tau_w$), the friction velocity ($U_\tau$), and Equations (18) and (19), respectively, by considering the closest distance between the cell and the wall, it is possible to replace Equation (16) and obtain a range of values.

$$C_f = 0.078 Re^{-0.25} \dots \tag{17}$$

$$\tau_w = 0.5 C_f \rho U_\infty^2 \dots \tag{18}$$

$$U_\tau = \sqrt{\frac{\tau_w}{\rho}} \dots \tag{19}$$

Finally, there is a range y+, which belongs to the viscous sublayer region (blue circle of Figure 7), that is used. The wall option "ENHANCED WALL TREATMENT" function is recommended for (**y$^+$** < 5) [20], and for the convergence criterion, it uses the k-epsilon RNG turbulence model [21]; this gives the energy equation and the multiphase "Volume of Fluid" (VOF) model [22].

In Table 5, the features of each generated mesh are depicted, indicating their y+, node count, cell count, and the time required to complete 10,000 iterations. The numerical simulations were conducted using an Intel Core i7 processor on an HP Envy Leap 17.

Multiphase is mentioned according to the VOF approach, which may exist in liquid-gas or liquid-liquid interactions, with our case being a type of separate system; in other words, it is a continuous flow, such as in the case of water with oil. It is called a phase if it is in a state or if it is the place it occupies (dispersed or continuous), as in the experiments carried out using VOF simulation for single rising drops in three liquid-liquid extraction systems using csf and css interfacial force models [23]; this included water and n-butanol,

water—n.butyl acetate, and water—toluene, where water constituted the continuous phase in all cases, and the other components constituted the dispersed phase [24].

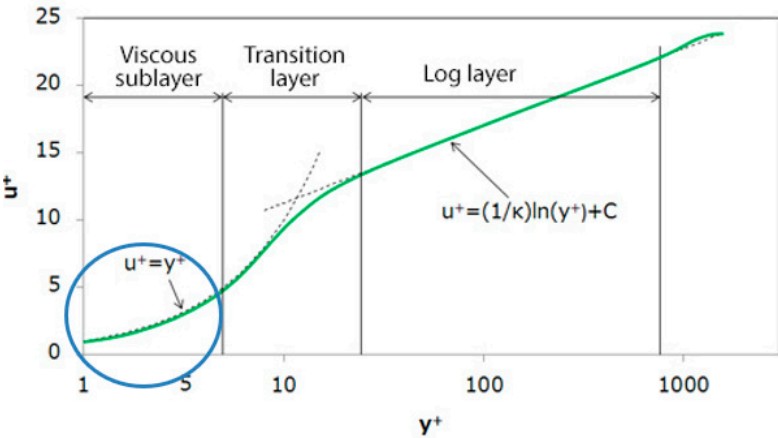

**Figure 7.** Velocity distribution [16].

**Table 5.** Characteristics of the mesh.

| Characteristics | Tetrahedral | Hexahedral |
|---|---|---|
| y (mm) | $3.2 \times 10^{-3}$ | $4 \times 10^{-3}$ |
| y+ | 3.002 | 3.655 |
| Cells | 266,119 | 6,048,104 |
| Faces | 617,024 | 18,265,032 |
| Nodes | 99,037 | 6,169,490 |
| 10,000 Iterations Time (hours) | 20 | 42 |

It is important to note that the phases involved in the simulation are air, nitrogen, and water (Equation (20)) [25]. Initially, the numerical simulation was conducted for each atomizer independently. Each atomizer was subjected to five different inlet conditions: ΔP = 150, 250, 350, 450, and 550 kPa. Additionally, a numerical simulation was performed with both injectors in operation, using ΔP = 350 kPa for the nitrogen atomizer and ΔP = 350 kPa for the water atomizer. The initial temperatures were set to 4.85 °C and −190.15 °C for the water and nitrogen atomizers, respectively.

$$\frac{\partial}{\partial t}(\eta\rho\Phi)_m + \frac{\partial}{\partial x_i}(\eta\rho u_i\Phi)_m = \frac{\partial}{\partial x_i}\left(\Gamma_\Phi \frac{\partial\Phi}{\partial x_i}\right)_m + (\eta S_\Phi)_m + \sum_{n=1}^{N_p} C_{\Phi,\mathrm{mn}}(\Phi_n - \Phi_m)$$
$$+m \sum_{n=1}^{N_p} C_{\Phi,\mathrm{mn}}\left(\dot{m}_{mn}\Phi_n - \dot{m}_{mn}\Phi_m\right) \ldots \tag{20}$$

where

$\eta$: Volumetric fraction of fluid.
$\Phi$: Scalar.
$\rho$: Density.
$u_i$: Velocity component in the i direction.
$\Gamma_\Phi$: Diffusion coefficient for a scalar $\Phi$.
$S_\Phi$: Source term for a scalar $\Phi$.
$N_p$: Nú Total number of phases.
$C_{\Phi,\mathrm{mn}}$: Mass transfer coefficient between phases m and n.
$\dot{m}_{mn}$: Mass variation per unit volume of phase m to phase n.

In the context of computational fluid dynamics (CFDs), ANSYS ICEM CFD emerges as a sophisticated and specialized mesh generation tool. Meticulously crafted to address the intricacies of complex geometries and nuanced simulations, ICEM CFD empowers users with the capacity to forge high-quality meshes. The inherent flexibility of this tool enables

a seamless adaptation to intricate geometrical configurations, while its advanced control mechanisms facilitate precise adjustments, thereby ensuring meticulous alignment between simulation objectives and mesh refinement. In contrast, within the intricate tapestry of ANSYS simulation solutions, ANSYS Workbench Meshing assumes a pivotal role as an integral component, presenting an accessible pathway to effective mesh generation. This tool streamlines the process through the provision of automation and integration within the ANSYS Workbench environment. It caters to users aiming for a smooth transition from geometry to analysis while harnessing automation to expedite mesh creation. The simplified approach of Workbench Meshing caters to users who prioritize efficiency and a user-friendly interface in consonance with the pursuit of prompt and dependable simulation outcomes. In the context of this project, both options were considered, ultimately leading to the selection of ANSYS ICEM CFD as the preferred choice [26].

For this project, we used RANS models (Reynolds-Averaged Navier-Stokes), which constitute a category of mathematical models employed in the numerical simulation of turbulent flows within fluid mechanics. These models are predicated on decomposing flow variables into a mean component (temporally averaged) and a fluctuating component. These models find extensive utilization in addressing turbulent flow challenges across engineering and scientific research domains. Through numerically solving time-averaged Navier-Stokes equations, RANS models facilitate the estimation of statistical flow features, such as mean velocity and concentration profiles [27,28]. A clear example of the Rans Method is the journal A Study of RANS Turbulence Models in Fully Turbulent Jets: A Perspective for CFD-DEM Simulations, in which the authors investigate RANS turbulence models in fully turbulent jets, and their relevance for CFD-DEM simulations focus on understanding computational fluid dynamics (CFDs) with simulations of discrete particle mechanics. These methods allow for the prediction of the behavior of turbulent flow in specific conditions. Where CFDs is used to analyze and predict the behavior of the fluid flow in a predetermined domain [29].

Computational fluid dynamics (CFDs) simulations can prove exceptionally valuable in the design and optimization of water/nitrogen atomizers or other spray systems. Here are several ways in which CFDs can contribute to the design process [30,31]:

- Geometry design: CFDs enables the evaluation of different atomizer designs prior to physical fabrication.
- Fluid-structure interaction: CFDs can be used to assess how the flow of a liquid and gas affects the atomizer's structure and vice versa.
- Virtual experiment design: CFDs enables virtual experiments under different operating conditions before physical implementation.
- Interaction analysis with the environment: CFDs can simulate how atomized droplets interact with surfaces, air currents, or other fluids in the environment.

Figure 8 shows a flowchart in which the summarized steps are indicated in order to reach the expected results of the Project, for which, as explained above, in the creation of geometries, it is necessary to define the initial parameters and use reverse engineering, and with the ideal method and viscosity losses, it is possible to obtain the geometry that satisfies the operation of the atomizer; then, we proceed to the mesh elaboration, and once in Fluent, the boundary conditions and interactions are chosen for the fluids; when this is simulated if it does not converge, the mesh must be improved, and the same process must be carried out; if it does converge, the final results are observed.

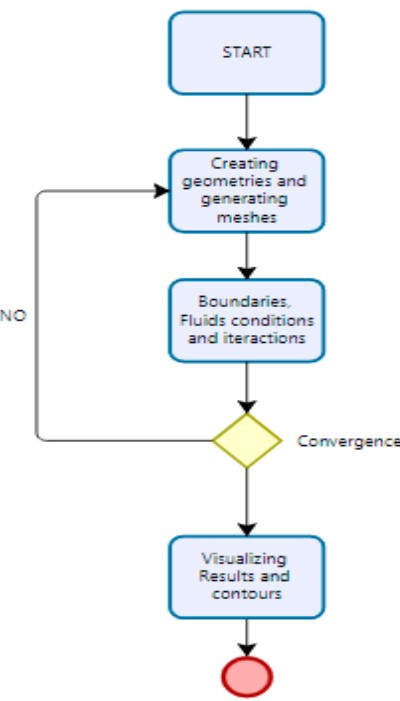

**Figure 8.** Flowchart.

## 4. Results

During this project phase, the outcomes attained through the Fluent software are subject to analysis. In both mesh types (tetrahedral and hexahedral), the findings are presented in Tables 6 and 7, encompassing the parameters, such as the mass flow attained under varying inlet pressures, for both nitrogen and water.

**Table 6.** Mass flow as a function of a drop in pressure from $LN_2$.

| | LIQUID NITROGEN | | | | | | |
|---|---|---|---|---|---|---|---|
| | Mass Flow, (kg/s) | | | | | | |
| Differential Pressure, (ΔP/kPa) | Mathematical Model | Simulation Tetrahedral | Deviation Mathematical Model Respect to Mesh Tetrahedral (%) | Simulation Hexahedral | Deviation Mathematical Model Respect to Mesh Hexahedral (%) | Heat Transfer Rate (KW) Tetrahedral | Heat Transfer Rate (KW) Hexahedral |
| 150 | 0.0229 | 0.0252 | −10.0437 | 0.0259 | −13.1004 | −11.076 | −11.373 |
| 250 | 0.0296 | 0.0324 | −9.4595 | 0.0332 | −12.1622 | −14.241 | −14.578 |
| 350 | 0.035 | 0.0382 | −9.1429 | 0.0392 | −12.0000 | −16.791 | −17.213 |
| 450 | 0.0397 | 0.0432 | −8.8161 | 0.0443 | −11.5869 | −18.988 | −19.453 |
| 550 | 0.0439 | 0.0478 | −8.8838 | 0.0491 | −11.8451 | −21.011 | −21.560 |

**Table 7.** Mass flow as a function of a drop in pressure from $H_2O$.

| | WATER | | | | | | |
|---|---|---|---|---|---|---|---|
| | Mass Flow, (kg/s) | | | | | | |
| Differential Pressure, (ΔP/kPa) | Mathematical Model | Simulation Tetrahedral | Deviation Mathematical Model Respect to Mesh Tetrahedral (%) | Simulation Hexahedral | Deviation Mathematical Model Respect to Mesh Hexahedral (%) | Heat Transfer Rate (KW) Tetrahedral | Heat Transfer Rate (KW) Hexahedral |
| 150 | 0.1600 | 0.1631 | −1.9375 | 0.1721 | −7.5625 | −13.557 | −14.395 |
| 250 | 0.2070 | 0.2089 | −0.9179 | 0.2205 | −6.5217 | −17.364 | −18.444 |
| 350 | 0.2450 | 0.2482 | −1.3061 | 0.262 | −6.9388 | −20.631 | −21.915 |
| 450 | 0.2780 | 0.2789 | −0.3237 | 0.2944 | −5.8993 | −23.183 | −24.625 |
| 550 | 0.3070 | 0.3079 | −0.2932 | 0.325 | −5.8632 | −25.594 | −27.184 |

In Figure 9a,b, we can appreciate the curve formed by the results obtained in the simulation at different pressures and different mesh models for the mass flows. From the

figure, we can conclude that higher pressure leads to an increase in the mass flow. These results can be corroborated with the research by Liu X et al. [32], in which they state that higher pressure leads to an increase in mass flow based on their experimental study of pressure swirl atomizers with water and liquid nitrogen.

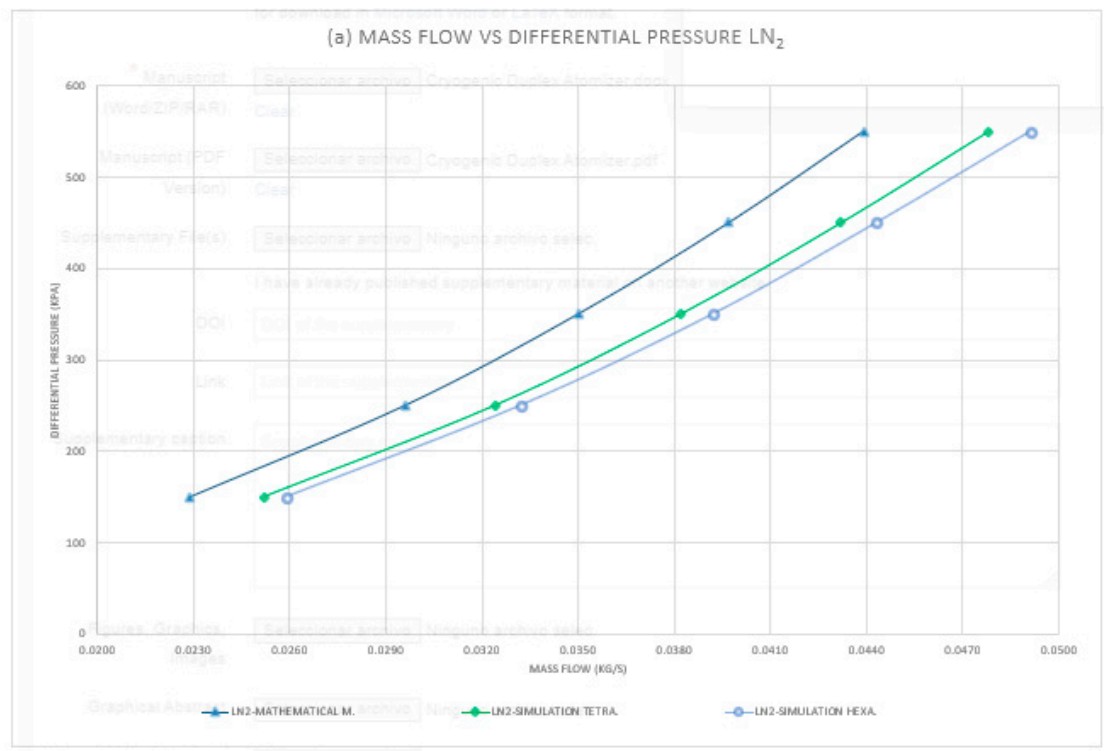

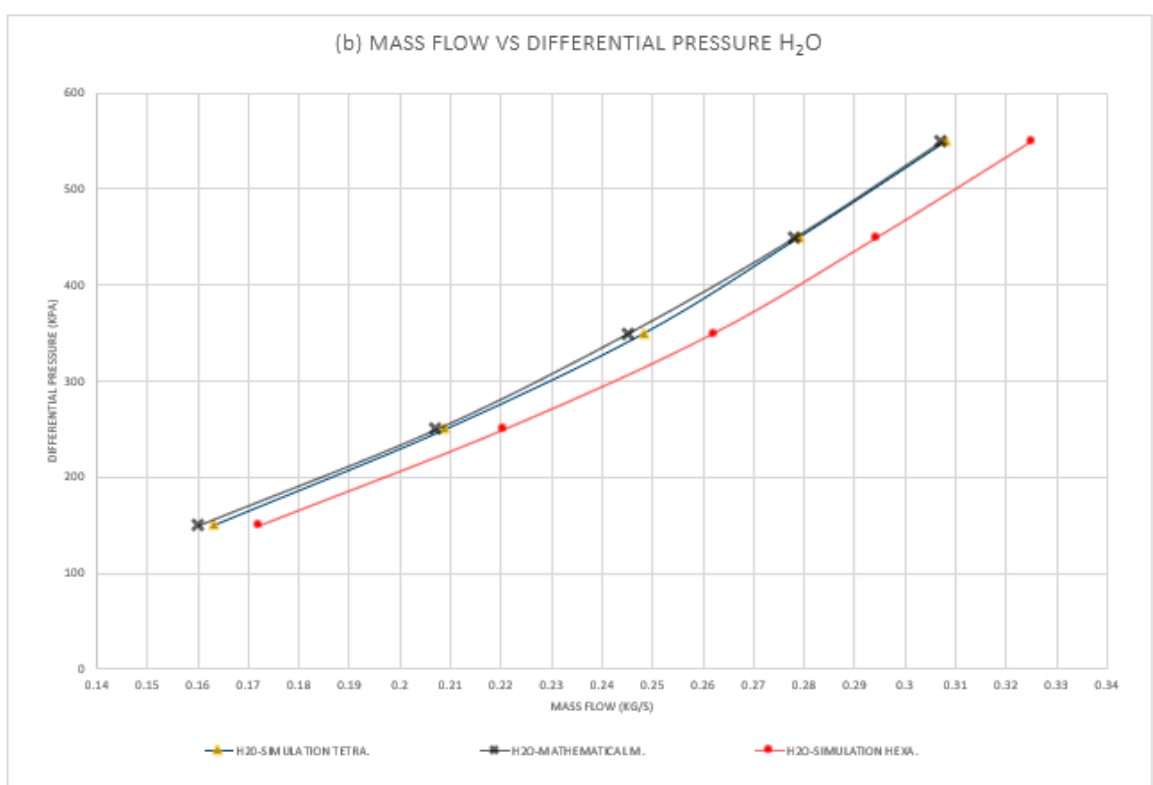

**Figure 9.** (**a**) Mass flow rate vs. pressure: liquid nitrogen; (**b**) mass flow rate vs. pressure: water.

In Figure 10, we can observe the density contours of the involved fluids, such as water (red color), liquid nitrogen (yellow color), and air (blue color). We can see the phase change of the liquid nitrogen when it interacts with air and water at the atomizer outlet, as well as the conical shape of the resulting sprays in the hexahedral model.

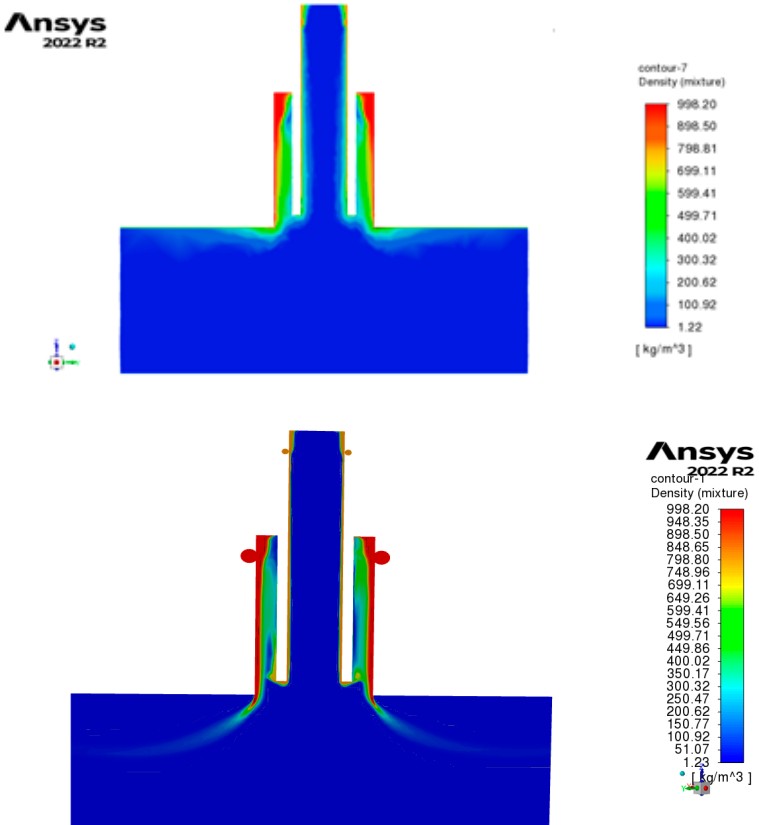

**Figure 10.** Density contour of duplex pressure-swirl atomizer (nitrogen atomizer: 350 Kpa; water atomizer: 350 Kpa) with tetrahedral mesh (**upper side**) and hexahedral mesh (**lower side**).

In Figure 11, we can observe the low pressures recorded in the "air core" of both the nitrogen and water atomizers, which results in high-speed air re-entry into the swirl chambers (approximately 10 to 17 m/s), as can be seen in the velocity vectors in Figure 12.

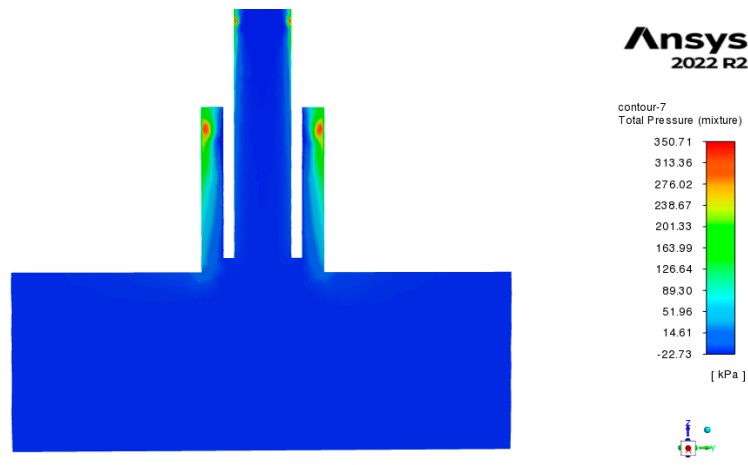

**Figure 11.** *Cont.*

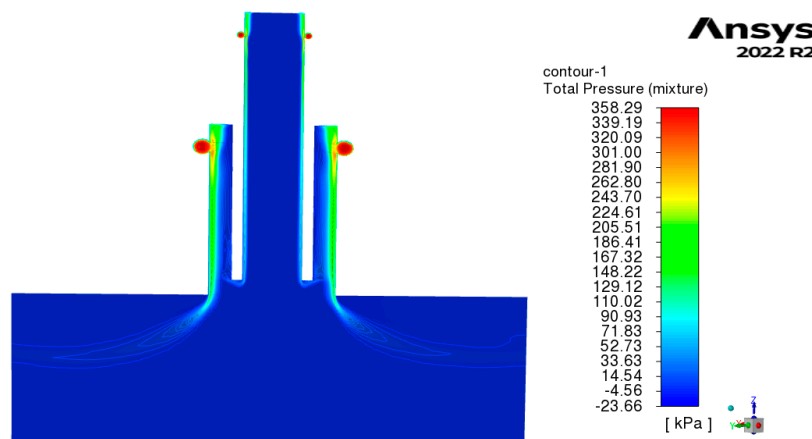

**Figure 11.** Total pressure contour of duplex pressure swirl atomizer (nitrogen atomizer: 350 Kpa; water atomizer: 350 Kpa) with tetrahedral mesh (**upper side**) and hexahedral mesh (**lower side**).

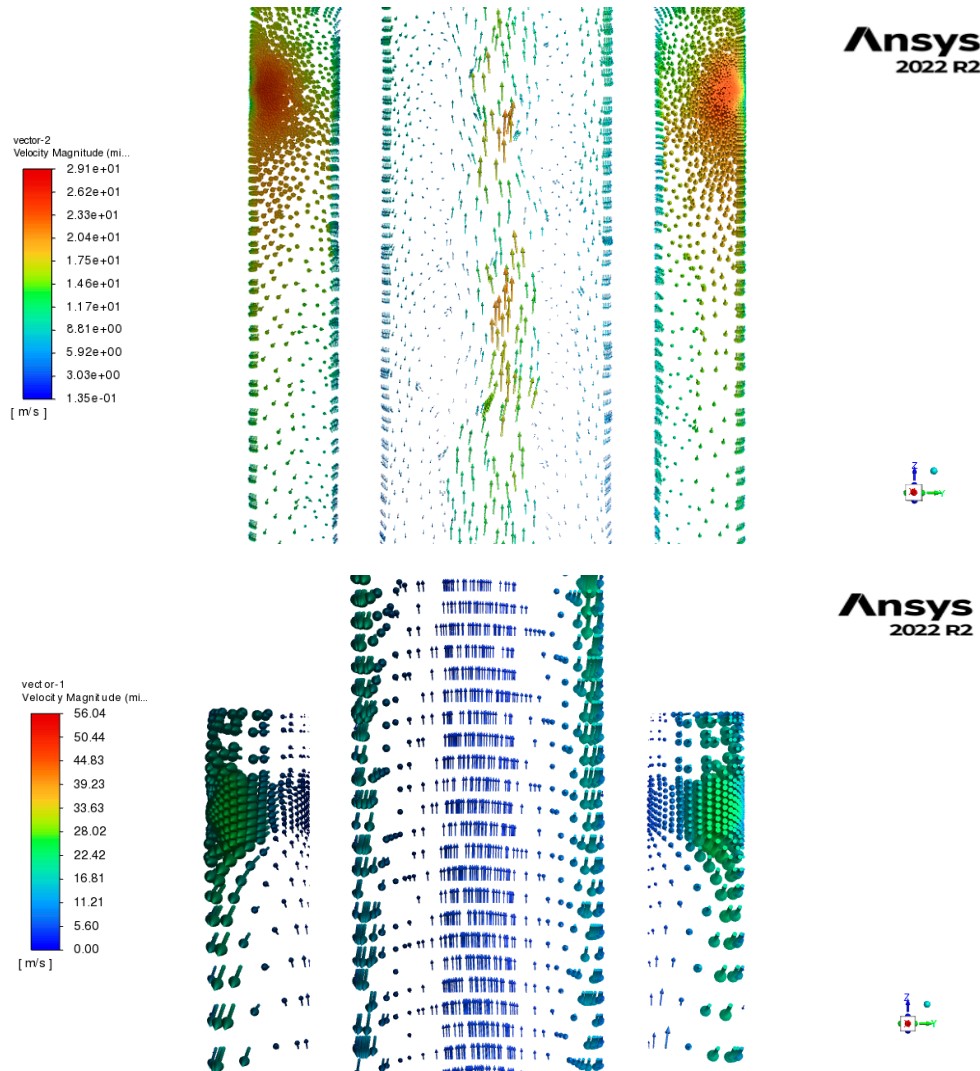

**Figure 12.** Velocity vectors of duplex pressure-swirl atomizer (nitrogen atomizer: 350 Kpa; water atomizer: 350 Kpa) with tetrahedral mesh (**upper side**) and hexahedral mesh (**lower side**).

Finally, we will analyze Figure 13, which represents the temperature contour. It should be mentioned that the initial temperatures were 4.85 °C and −190 °C for the water

and nitrogen atomizers, respectively, through the tangential channels. In Figure 11, we can observe three important regions: A, B, and C. Region A is within a temperature range of [−16.56 °C; 12.38 °C], where we can see that the water film is still in a liquid state (approximately 12.38 °C) and its temperature rapidly decreases upon collision with nitrogen at the atomizer outlet, reaching a temperature of −16.56 °C. This guarantees that the phase change of water from liquid to solid will occur soon (water freezing point: 0 °C), which is the objective of this work. On the other hand, in region B, we can observe a temperature range of [−190.18 °C; −146.78 °C], indicating that this region will always remain at cryogenic temperatures, ensuring the rapid cooling of any nearby fluid. Finally, in region C, we can see the temperature behavior of both fluids after being expelled from the atomizer, within a range of [−16.56 °C; 2.09 °C], demonstrating that the cold environment generated by liquid nitrogen has a relevant effectiveness, achieving the objective of lowering the water temperature and (experimentally) expecting hail. In addition, the development of the hexahedral mesh helps to observe higher quality and better interaction of the phases.

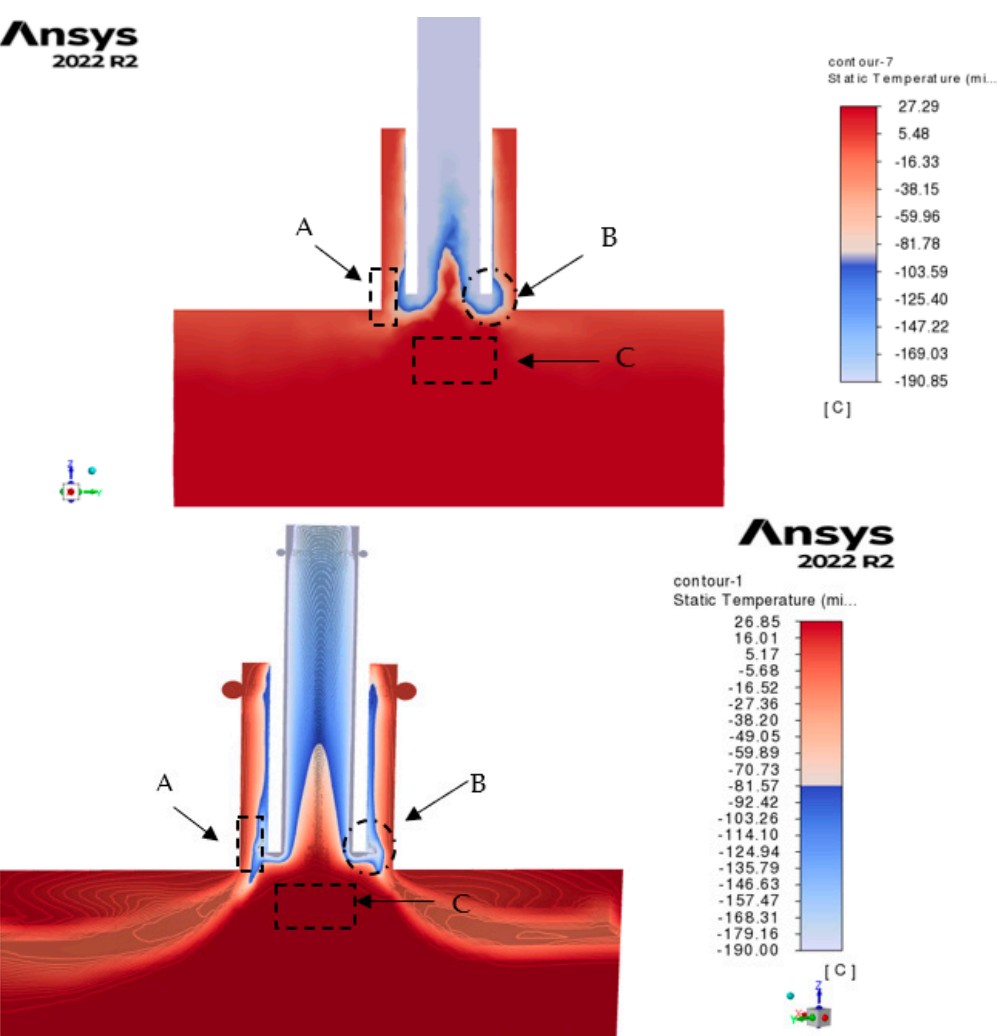

**Figure 13.** Temperature contour of duplex pressure-swirl atomizer (nitrogen atomizer: 350 Kpa; water atomizer: 350 Kpa) with tetrahedral mesh (**upper side**) and hexahedral mesh (**lower side**).

## 5. Conclusions

The mass flow rate as a function of the pressure differential (ΔP) is important as a starting point for the design of atomizers. Therefore, we sought to verify if the values from the mathematical model could be validated with the values from the numerical simulation, as shown in the curves of Figure 9a,b, demonstrating such approximation (with a maximum

error of 1.94% for the water atomizer and 10% for the LN$_2$ atomizer), which ensures that the final dimensions of the atomizers are correct for manufacturing. It is worth noting that the models used in the numerical simulation, such as the RNG k-$\varepsilon$ turbulence model and the Volume of Fluid (VoF) multiphase model, were crucial in understanding the flow behavior both inside and outside these atomizers. This led to the recording of the temperature of water when interacting with liquid nitrogen, ensuring the solidification of water droplets at the atomizer outlet, which is the purpose of this work related to the cold preservation of seafood products. This is corroborated by the research "Cryogenic fluid dynamics of pressure swirl injectors at supercritical conditions" [33], where it can be observed that at the outlet of these types of atomizers, the cryogenic fluid atomization has a large volume range at low temperatures, which can be used for the cooling of food products and metallurgical treatments. The implementation of the hexahedral mesh reveals an enhanced simulation interface and phase interaction. Furthermore, upon comparing both mesh models, in the case of the tetrahedral model, it incurs lower computational consumption and exhibits a reduced interface quality; however, the mass flow values closely approximate those of the mathematical model. Conversely, within the hexahedral model, higher computational resources are utilized due to the greater number of elements. Nevertheless, the mass flow values also approximate the mathematical model, and the interface quality and result visualization attain an excellent standard. Consequently, the hexahedral mesh model emerges as the optimal choice and is recommended for projects of this nature.

**Author Contributions:** Conceptualization, E.A. and D.R.; methodology, E.A. and D.R.; software, D.R.; validation, J.R.; writing—review and editing, J.R. and N.V.; visualization, N.V.; supervision J.R., N.V. and G.R. All authors have read and agreed to the published version of the manuscript.

**Funding:** This research was funded by direction of the Peruvian University of Applied Sciences grant number EXPOST-2023-2 And The APC was funded by 1600.

**Data Availability Statement:** Data available upon request due to privacy and ethics restrictions. The data presented in this study are available upon request from the corresponding author. The data are not publicly available due to the sensitivity of the data and information created and analyzed in this study, the models it can also be found of bibliographic sources.

**Acknowledgments:** The authors of this research wish to express their gratitude to the UPC (Universidad Peruana de Ciencias Aplicadas) and its research department for their guidance and support during the duration of this study and its financing through the UPC-EXPOST-2023-2 incentive. which led to the development of this project.

**Conflicts of Interest:** The authors declare no conflict of interest.

## Nomenclature

| | |
|---|---|
| A | geometrical characteristics parameter of pressure swirl atomizer of tangential inlets |
| $A_E$ | equivalent geometrical characteristics parameter due to the viscosity of swirl atomizers |
| $A_c$ | geometrical characteristics parameter of conical pressure swirl atomizer |
| Cd | discharge coefficient |
| $f_p$ | cross-sectional area of inlet port |
| K | coefficient of loss due to liquid viscosity |
| n | number of inlet channels |
| $\dot{m}$ | mass flow rate |
| $\Delta P$ | differential pressure |
| Re | Reynolds number |
| $R_s$ | swirl chamber radius |
| $R_{inj}$ | radius to axis inlet channel |
| $r_a$ | air core radius |
| $r_o$ | outlet orifice radius |
| u | vectorial velocity |
| $U_{in}$ | inlet entrance velocity |
| U,W | velocities |

**Greek Symbols**

α    half-spray angle
ξ    losses Coefficient
$\eta$    volume fraction of fluid
φ    film flow area coefficient
Φ    scalar
λ    resistance coefficient of Blasius
μ    liquid absolute viscosity
ρ    liquid density
σ    liquid surface tension
$\upsilon$    liquid kinemátic viscosity

**Subscripts**

a    air core
eq    equivalent parameter due to viscosity
inj    parameters related to inlet channels.
liq    liquid
r    radial component
s    swirl chamber.
tot    total
θ    tangential component
z    axial component

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
