# Peer review of "Design of a Cryogenic Duplex Pressure-Swirl Atomizer through CFDs for the Cold Conservation of Marine Products"

_fluids, doi:10.3390/fluids8100271_

Round 1
Reviewer 1 Report
Please read the attachment. Thank you.

Major changes in grammar and structure are needed.
Author Response
The file is uploaded:
Article with comments attached with the answers to the constructive questions

Reviewer 2 Report
The present paper is about the instantaneous ice preparation by mixing the water and LN2. The concept is novel. However, the manuscript need to be revised thoroughly before accepting for the publication.
1. In the abstract it is mentioned that “atomized water to cause heat loss and bring it to its melting point”. It should be freezing point.
2. How the atomizer parameters (Table 1) have chosen not explained in the manuscript.
3. Before discussing the dimensions (Table 1), it is required to discuss the geometry first.
4. Figure 1 not cited in the paper
5. In many cases improper subscripts are used which need to be corrected, Ex: Table 4, H2O should be H2O and also N2 should be N2.
6. Grid independence study is not reported in the manuscript.
7. In the line 216, (Page 8), air, water and nitrogen are mentioned as the phases. But, these are three different materials, but not phases. When flowing in the nozzle these are changing the phase. Hence, it is genuine to mention three phases, gaseous, liquid and solid phases. This sentence to be revised and throughout the manuscript.
8. There are 9 figures only in the manuscript. However, it is cited Fig. 10 in the line 274 (Page 11)
9. In the Eq 1, it should be Sin α (line 115).
10. The flow rates of Water, LN2 are given in the Table 5 and Table 6. However, the energy balance between water, Liquid nitrogen and heat losses are not reported in the manuscript.
11. Transient analysis may required to find the initial process before water converted into the ice.
12. In the Figure 7, the contours are shown for the material; air (blues-0), Water (green-1) and LN2 (Red-2). However, the caption of the figure is “phase contour…”. But it is not showing the phase change, like (ice, liquid and gas). The figure caption supposed to be changed and a new figure is required to add to explain the phase change.
Need to be improved
Author Response
the file is uploaded with raised comments highlighted in yellow

Reviewer 3 Report
The present work focuses on the performance of duplex atomizer based on numerical simulation. Overall, the works seems to be more technical report like. From the research point of view, there is no any innovation and academic contribution. Other specific comments are listed as below.
1: The mathematical model to be employed consists of two stages: first the fluid is considered as inviscid, and then in the viscosity losses are taken into account. Actaully, there is no new insight via such two stage analyses. Why the authors do this way?
2: Is Fig. 5 necessary? This standard curve is not from the simulation of the present work.
3: When I went through the simulation part, it is quite confusing. The simulation is RANS or LES? What is the turbulence model? How about the mesh?
Some eidting problems.
1: Line 287: , "demonstrating that the cold environment generated by liquid nitrogen has a significant reach" is not clear. What does it mean?
2: Writing needs to be checked carefully. In Eq. 18, there are some spelling typos, e.g. donde, Nú
3: Two Fig. 7s are shown
4: Quality of some figures are very bad, e.g. Fig. 7, Fig.8
Overall, technically the work in the present from is very weak and does not meet the standard of a professional journal. I will not recommend any consideration for publication.
English is basically OK.
Author Response
The file is uploaded with the lifted observations highlighted in yellow

Round 2
Reviewer 2 Report
The manuscript is edited appropriately as per the reviewer's comments. I want to recommend the paper to accept.